# Formulation and Development of Bioadhesive Oral Films Containing *Usnea barbata* (L.) F.H.Wigg Dry Ethanol Extract (F-UBE-HPC) with Antimicrobial and Anticancer Properties for Potential Use in Oral Cancer Complementary Therapy

**DOI:** 10.3390/pharmaceutics14091808

**Published:** 2022-08-28

**Authors:** Violeta Popovici, Elena Matei, Georgeta-Camelia Cozaru, Laura Bucur, Cerasela Elena Gîrd, Verginica Schröder, Emma Adriana Ozon, Iulian Sarbu, Adina Magdalena Musuc, Irina Atkinson, Adriana Rusu, Simona Petrescu, Raul-Augustin Mitran, Mihai Anastasescu, Aureliana Caraiane, Dumitru Lupuliasa, Mariana Aschie, Victoria Badea

**Affiliations:** 1Department of Microbiology and Immunology, Faculty of Dental Medicine, Ovidius University of Constanta, 7 Ilarie Voronca Street, 900684 Constanta, Romania; 2Center for Research and Development of the Morphological and Genetic Studies of Malignant Pathology, Ovidius University of Constanta, CEDMOG, 145 Tomis Blvd., 900591 Constanta, Romania; 3Clinical Service of Pathology, Sf. Apostol Andrei Emergency County Hospital, 145 Tomis Blvd., 900591 Constanta, Romania; 4Department of Pharmacognosy, Faculty of Pharmacy, Ovidius University of Constanta, 6 Capitan Al. Serbanescu Street, 900001 Constanta, Romania; 5Department of Pharmacognosy, Phytochemistry, and Phytotherapy, Faculty of Pharmacy, Carol Davila University of Medicine and Pharmacy, 6 Traian Vuia Street, 020956 Bucharest, Romania; 6Department of Cellular and Molecular Biology, Faculty of Pharmacy, Ovidius University of Constanta, 6 Capitan Al. Serbanescu Street, 900001 Constanta, Romania; 7Department of Pharmaceutical Technology and Biopharmacy, Faculty of Pharmacy, Carol Davila University of Medicine and Pharmacy, 6 Traian Vuia Street, 020956 Bucharest, Romania; 8Department of Pharmaceutical Physics and Biophysics, Drug Industry and Pharmaceutical Biotechnologies, Faculty of Pharmacy, “Titu Maiorescu” University, 004051 Bucharest, Romania; 9“Ilie Murgulescu” Institute of Physical Chemistry, Romanian Academy, 202 Spl. Independentei, 060021 Bucharest, Romania; 10Department of Oral Rehabilitation, Faculty of Dental Medicine, Ovidius University of Constanta, 7 Ilarie Voronca Street, 900684 Constanta, Romania

**Keywords:** *Usnea barbata* (L.) F. H. Wigg dry ethanol extract, oral squamous cell carcinoma, bioadhesive oral films, usnic acid, antimicrobial activity, CLS-354 cell line, blood cell cultures, oxidative stress, anticancer potential

## Abstract

Medical research explores plant extracts’ properties to obtain potential anticancer drugs. The present study aims to formulate, develop, and characterize the bioadhesive oral films containing *Usnea barbata* (L.) dry ethanol extract (F-UBE-HPC) and to investigate their anticancer potential for possible use in oral cancer therapy. The physicochemical and morphological properties of the bioadhesive oral films were analyzed through Fourier transform infrared spectroscopy (FTIR), scanning electron microscopy (SEM), Atomic Force Microscopy (AFM), thermogravimetric analysis (TG), and X-ray diffraction techniques. Pharmacotechnical evaluation (consisting of the measurement of the specific parameters: weight uniformity, thickness, folding endurance, tensile strength, elongation, moisture content, pH, disintegration time, swelling rate, and ex vivo mucoadhesion time) completed the bioadhesive films’ analysis. Next, oxidative stress, caspase 3/7 activity, nuclear condensation, lysosomal activity, and DNA synthesis induced by F-UBE-HPC in normal blood cell cultures and oral epithelial squamous cell carcinoma (CLS-354) cell line and its influence on both cell types’ division and proliferation was evaluated. The results reveal that each F-UBE-HPC contains 0.330 mg dry extract with a usnic acid (UA) content of 0.036 mg. The bioadhesive oral films are thin (0.093 ± 0.002 mm), reveal a neutral pH (7.10 ± 0.02), a disintegration time of 118 ± 3.16 s, an ex vivo bioadhesion time of 98 ± 3.58 min, and show a swelling ratio after 6 h of 289 ± 5.82%, being suitable for application on the oral mucosa. They displayed in vitro anticancer activity on CLS-354 tumor cells. By considerably increasing cellular oxidative stress and caspase 3/7 activity, they triggered apoptotic processes in oral cancer cells, inducing high levels of nuclear condensation and lysosomal activity, cell cycle arrest in G0/G1, and blocking DNA synthesis. All these properties lead to considering the UBE-loaded bioadhesive oral films suitable for potential application as a complementary therapy in oral cancer.

## 1. Introduction

Oral cancer belongs to the head and neck cancers group, which includes the larynx, throat, lips, mouth, nose, and salivary glands malignancies [1]. It is the sixth most common neoplastic disease worldwide [2]. Tobacco or/and alcohol use, betel-quid chewing, and human papillomavirus (HPV) infection are adequately evidenced as risk factors associated with oral cancer [3]. These confirmed factors, associated with pollution, diet and nutrition, poor oral hygiene, chronic inflammation, infectious diseases (i.e., with *C. albicans* [4]), and a hereditary predisposition, have contributed to the increased risk of developing oral cavity malignancies [5]. The conventional treatment of oral cancer consists of surgical intervention and radiotherapy; in an advanced stage, they could be combined with chemotherapy after clinical trials. Radiation exposure makes oral tissues more easily damaged, and physiological repair mechanisms are compromised due to permanent cellular damage [6]; thus, radiotherapy-induced oral complications occur (i.e., oral mucositis, bacterial and fungal infections, saliva change, mucosal fibrosis, sensory dysfunctions, dental caries, periodontal disease, and osteoradionecrosis) [6,7]. The optimal combination with high benefits for oral cancer patients is still unknown, and 5-year survival is around 52% [3] due to its invasive character and late diagnosis.

The National Cancer Institute also accepts non-conventional products and practices of Complementary and Alternative Medicine (CAM) https://www.cancer.gov/about-cancer/treatment/cam/patient (accessed on 30 July 2022). Numerous conventional medical care practitioners are also CAM ones [8] and support global research investigating plant extracts’ properties to obtain potential anticancer drugs. The rate of patients using CAM in addition to the standard therapy regime is 40–90% [9], and plant-based products could help them to prevent and recover from oral complications.

Many in vitro, in vivo, and clinical studies proved the benefits of various plant-based formulations for oral cancer patients. Thus, a recently published review highlights the promising effects of milk thistle extracts in oral pathology, i.e., oral cancer, oral mucositis (OM), periodontal disease, dental caries, and oral candidiasis [10]. Other authors evidenced topically applied chamomile’s effectiveness in OM—the most painful acute oral complication of radio and chemotherapy [11,12]. A previous clinical study revealed the benefits of SAMITAL^®^ (Indena S.p.A.), a phytopharmaceutical combination of three highly standardized botanical drug extracts *(Vaccinium myrtillus* fruits, *Macleaya cordata* fruits, and *Echinacea angustifolia* roots) in OM treatment [13].

Choi et al. [14] recommend the mixture of *Momordica charantia* Linn., *Pistacia lentiscus*, and *Commiphora myrrha* supercritical extracts as a preventive and therapeutic agent for oral mucosa inflammation and oral cancer, supported by an in vitro and in vivo evaluation. The present study suggests *U. barbata* dry ethanol extract for the previously mentioned applications, based on animal model cytotoxicity prescreen and in vitro studies. Moreover, recent studies proposed pyroligneous extracts films of *Eucalyptus grandis* and chitosan [15] and mucoadhesive buccal films with *Aloe barbadensis* dry extract [16] for oral cancer prevention and treatment. In this work, *U. barbata* (L.) F.H. Wigg dry ethanol extract (UBE) was incorporated in bioadhesive oral films. Then, UBE-loaded films were analyzed through complex physicochemical, pharmacotechnical, and biological studies, aiming to investigate their suitability for oral administration and pharmacological activities for potential application as a complementary therapy in oral cancer.

## 2. Materials and Methods

### 2.1. Materials

This study’s chemicals, reagents, and standards were of analytical grade. Usnic acid standard 98.1% purity, Propidium Iodide (PI) 1.0 mg/mL, Dimethyl sulfoxide (DMSO), Polyethylene Glycol 400 (PEG 400), and Hydroxypropyl cellulose (HPC) and Antibiotics mix solution—100 µL/mL with 10 mg Streptomycin, 10,000 U Penicillin, 25 µg Amphotericin B per 1 mL—were provided by Sigma-Aldrich Chemie GmbH (Taufkirchen, Germany). Annexin V Apoptosis Detection Kit and flow cytometry staining buffer (FCB) were purchased from eBioscience^TM^ (Frankfurt am Main, Germany) and RNase A 4 mg/mL from Promega (Madison, WI, USA). Magic Red^®^ Caspase-3/7 Assay Kit, Reactive Oxygen Species (ROS) Detection Assay Kit, and EdU i-Fluor 488 Kit were supplied by Abcam (Cambridge, UK).

The OSCC cell line (CLS-354) and the culture medium—Dulbecco’s Modified Eagle’s Medium (DMEM) High Glucose, basic supplemented with 4.5 g/L glucose, L-glutamine, and 10% Fetal Bovine Serum (FBS) were provided by CLS Cell Lines Service GmbH (Eppelheim, Germany). Trypsin–ethylenediamine tetra acetic acid (Trypsin EDTA) and the media for blood cells—Dulbecco’s phosphate-buffered saline with MgCl_2_ and CaCl_2_, FBS, and L-Glutamine (200 mM) solution—were purchased from Gibco^TM^ Inc. (Billings, MT, USA).

The blood samples were collected from a non-smoker healthy donor (B Rh+ blood type), according to Ovidius University of Constanta Ethical approval code 7080/10.06.2021 and Donor Consent code 39/30.06.2021.

*U. barbata* lichen was harvested in March 2021 from the branches of conifers in the forest localized in the Călimani Mountains (47°29′ N, 25°12′ E, and 900 m altitude). It was identified by the Department of Pharmaceutical Botany of the Faculty of Pharmacy, Ovidius University of Constanta, using standard methods. A voucher specimen is deposited in the Herbarium of Pharmacognosy Department, Faculty of Pharmacy, Ovidius University of Constanta (Popovici 3/2021, Ph-UOC). The 96% ethanol for *U. barbata* dry extract preparation was provided by Chimreactiv SRL Bucharest, Romania.

*Artemia salina* eggs and *Artemia* salt (Dohse Aquaristik GmbH & Co., Gelsdorf, Germany) were purchased online from https://www.aquaristikshop.com/ (accessed on 5 May 2022).

Bacterial and fungal cell lines (*S. aureus* ATCC 25923, *P. aeruginosa* ATCC 27353, *C. albicans* ATCC 10231, and *C. parapsilosis* ATCC 22019) for antimicrobial activity evaluation were obtained from Microbiology Department, S.C. Synevo Romania S.R.L., Constanta Laboratory, in partnership agreement No 1060/25.01.2018 with the Faculty of Pharmacy, Ovidius University of Constanta. Culture medium Mueller-Hinton agar (MHA) was supplied by Thermo Fisher Scientific, GmbH, Dreieich, Germany; RPMI 1640 Medium and Resazurin solution (from In Vitro Toxicology Assay Kit, TOX8-1KT, Resazurin based) were purchased from Sigma-Aldrich Chemie GmbH (Taufkirchen, Germany).

### 2.2. The Development of UBE-Loaded Bioadhesive Oral Films

Bioadhesive films with the same excipients but no active ingredient load were obtained and utilized as References to show the lichen extract’s efficacy and influence on the pharmacotechnical properties.

The composition of the developed bioadhesive films is displayed in Table 1.

The UBE dosage was established considering its solubility in ethanol (3% *w*/*w*). HPC was dispersed in water using a Heidolph MR 3001K magnetic stirrer at 750 rpm and room temperature; then, PEG 400 was added and blended. UBE was dissolved in ethanol, and the obtained solution was slowly poured into the polymer matrix while stirring under the same conditions. The final hydrogels were left overnight at room temperature for deaeration, then poured into Petri glass plates in a thin layer and let to dry for 24 h at ambient temperature. Dry films were peeled away from the plate surface and cut into 1.5 cm × 2 cm patches.

### 2.3. Physico-Chemical Characterization of the UBE-Loaded Bioadhesive Oral Films

#### 2.3.1. SEM Analysis

A high-resolution scanning electron microscope Quanta3D FEG (Thermo Fisher Scientific, GmbH, Dreieich, Germany) was used to evaluate both films (R and F-UBE-HPC) morphology.

#### 2.3.2. Atomic Force Microscopy (AFM)

AFM investigations were performed in non-contact mode, using decoupled, flexure-guided crosstalk eliminated scanners with an XE-100 microscope from Park Systems. In all AFM measurements, sharp tips were used, NCHR from Nanosensors^TM^, with typically ~8 nm radius of curvature, ~125 µm mean length, 30 µm mean width, ~42 N/m force constant, and ~330 kHz resonance frequency. XEI program (v 1.8.0) from Park Systems was used for image processing and roughness evaluation. Below the AFM images, presented as “enhanced contrast” view mode, representative line scans are plotted, showing in detail the surface profile of the scanned samples.

#### 2.3.3. Fourier Transform Infrared Spectroscopy (FTIR)

Bioadhesive oral films were characterized by Fourier Transform Infrared (FTIR) using a Nicolet Spectrometer 6700 FTIR and a Smart DuraSamplIR HATR (Horizontal Attenuated Total Reflectance) accessory with a laminated–diamond crystal (Thermo Electron Corporation, Waltham, MA, USA). FTIR spectra were achieved in the spectral range from 4000 to 400 cm^−1^, using a DTGS KBr detector, in transmittance mode. The spectra were the average of 32 scans recorded at 4 cm^−1^ resolution.

#### 2.3.4. Powder X-ray Diffractometry

The powder X-ray diffractometry (XRD) patterns were recorded using a Rigaku Ultima IV diffractometer (Rigaku Corporation, Tokyo, Japan) in parallel beam geometry. A Cu tube (λ = 1.54056 Å) operating at 40 kV voltage and 30 mA current was used. The scanning rate employed was 2°/min over a diffraction angle of 2θ, with a step size of 0.02 and a range of 5–60°.

#### 2.3.5. Thermogravimetric Analysis

The thermal curves of both bioadhesive oral films were recorded using a Mettler Toledo TGA/SDTA 851^e^ thermogravimetric analyzer (Mettler-Toledo GmbH, Greifensee, Switzerland). All experiments were conducted in a synthetic air atmosphere with a heating rate of 10 °C min^−1^ and a flow rate of 80 mL min^−1^.

### 2.4. Pharmacotechnical Properties of the UBE-Loaded Bioadhesive Oral Films

#### 2.4.1. Weight Uniformity

Twenty films from each series (F-UBE-HPC and R) were separately weighed, and the mean weight was calculated.

#### 2.4.2. Thickness

Thickness was measured on 20 films of each batch using a digital micrometer (Yato Trading CO., LTD, Shanghai, China) with a 0–25 mm measurement range and 0.001 mm resolution. The average values were calculated for both films.

#### 2.4.3. Folding Endurance

The films were folded and rolled repeatedly, at the same place, until they broke, or up to 300 times [17]. The folding times were recorded and reported as folding endurance values.

#### 2.4.4. Tensile Strength and Elongation Ability

A digital tensile force tester for universal materials was used to determine the tensile strength and elongation behavior of a Lloyd Instruments Ltd., LR 10K Plus, West Sussex, United Kingdom digital tensile force tester for universal materials. The analysis was carried out at a speed of 30 mm/min from a distance of 30 mm. The breaking force was measured by placing the patch in a vertical position between the two bracing. The tests were performed on five patches of each batch.

The tensile strength and elongation at break were calculated using the following equations:(1)$$\mathrm{Tensile}\;\mathrm{strength}\;\left(\mathrm{kg}/{\mathrm{mm}}^{2}\right)=\frac{\mathrm{Force}\;\mathrm{at}\;\mathrm{breakage}\;\left(\mathrm{kg}\right)}{\mathrm{Film}\;\mathrm{thickness}\;\left(\mathrm{mm}\right)\times \;\mathrm{Film}\;\mathrm{width}\;\left(\mathrm{mm}\right)}$$
(2)$$\mathrm{Elongation}\;\%\;=\frac{\mathrm{Increase}\;\mathrm{in}\;\mathrm{film}\;\mathrm{length}}{\mathrm{Inital}\;\mathrm{film}\;\mathrm{length}}\times 100$$

#### 2.4.5. Moisture Content

A thermogravimetric method was used to determine the drying loss using an HR 73 Mettler Toledo halogen humidity (Mettler-Toledo GmbH, Greifensee, Switzerland) [18]. Five films of each formulation were tested.

#### 2.4.6. Surface pH

Five films of each formulation were moistened for 5 min at room temperature with 1 mL distilled water (pH 6.5 ± 0.5). The pH value was registered by touching the film surface with the electrode of a CONSORT P601 pH-meter (Consort bvba, Turnhout, Belgium).

#### 2.4.7. In Vitro Disintegration Time

Using an Erweka DT 3 apparatus (Erweka^®^ GmbH, Langen, Germany), the time necessary for both films’ disintegration was determined in a simulated saliva phosphate buffer with a pH of 6.8 at 37 ± 2 °C [19].

#### 2.4.8. Swelling Ratio

Six films of each series were placed in Petri plates on a 1.5% agar gel and incubated at 37 ± 1 °C. The films were weighed every 30 min for 6 h. The swelling ratio is calculated through the following equation:(3)$$\mathrm{Swelling}\;\mathrm{ratio}\;=\frac{\mathrm{Wt}-\mathrm{Wi}}{\mathrm{Wi}}\times 100$$

Wt is the patch weight at time t after the incubation, and Wi is the initial weight [20,21,22].

#### 2.4.9. Ex Vivo Bioadhesion Time

This determination was assessed using the technique described by Gupta et al. [23] on a detached porcine oral mucosa. The fat layer and any tissue residue were removed from the membrane surface; then, it was fixed on a glass plate after being rinsed with distilled water and a phosphate buffer pH of 6.8 at 37 °C. Each film was hydrated in the center with 15 μL phosphate buffer and pressed onto the mucosa surface for 30 s. The glass plate was submerged in 200 mL phosphate buffer (pH 6.8) and kept at 37 °C for 2 min. A paddle set at a 28 rpm stirring rate simulated the oral cavity conditions, and the time requested for each film to detach from the oral mucosa was measured. This registered time is known as residence time or bioadhesion time. All tests were carried out in triplicate.

### 2.5. Antimicrobial Activity

#### 2.5.1. Inoculum Preparation

The direct colony suspension method (CLSI) was used to prepare the bacterial inoculum. Thus, bacterial colonies selected from a 24 h agar plate were suspended in M.H.A. medium, according to the 0.5 McFarland standard, measured at Densimat Densitometer (Biomerieux, Marcy-l’Étoile, France) with around 108 CFU/mL (CFU = colony-forming unit). The yeast inoculum was achieved using the same method, adjusting the RPMI 1640 with fungal colonies to the 1.0 McFarland standard, with 10^6^ CFU/mL.

#### 2.5.2. Samples and Standards

F-UBE-HPC was dissolved in 1 mL of diluted phosphate buffer. As standards, Ceftriaxone (Cefort 1g Antibiotice SA, Iași, Romania) solutions 30 mg/mL and 150 mg/mL in distilled water were used for bacteria. The Cefort powder was weighted at Partner Analytical balance (Fink & Partner GmbH, Goch, Germany) and dissolved in distilled water. Terbinafine solution 10.1 mg/mL (Rompharm Company S.R.L., Otopeni, România) was selected as standard for *Candida* sp.

#### 2.5.3. Microdilution Method

All successive steps were performed in an Aslair Vertical 700, laminar flow, microbiological protection cabinet (Asal Srl, Cernusco (MI) Italy). In four 96-well plates, we performed seven serial dilutions, adapting the protocol described by Fathi et al. [24].

All 96-well plates were incubated for 24 h at 37 °C for bacteria and 35 °C for yeasts in My Temp mini Z763322 Digital Incubator (Benchmark Scientific Inc., Sayreville, NJ, USA).

#### 2.5.4. Reading and Interpreting

The 96-well plates were examined with a free eye to see the color differences between standard and samples after 24 h incubation [25]. The corresponding sample concentration activities were compared with the standard antibiotic ones. For yeasts, the color chart of the resazurin dye reduction method was used [26,27].

### 2.6. Cytotoxic Activity of UBE-Loaded Bioadhesive Oral Films on A. salina Larvae

F-UBE-HPC was placed in 1 mL of diluted buffer and left to incubate for 15 min at 37 °C; after this time, its homogenous dispersion in the buffer solution was used as a treatment for *A. salina* larvae.

Brine shrimp larvae were achieved by placing the *A. salina* cysts in a saline solution of 0.35%, for 24–48 h, under continuous aeration and light, at room temperature. (20–22 °C). At the first larval stage, they were separated and introduced into experimental well plates (with a volume of 1 mL) in 0.3% saline solutions [28]. The prescreen was compared with a blank (untreated brine shrimp nauplii) to obtain accurate results regarding the F3 patch cytotoxic effect. During the test period, *A. salina* larvae were not fed to not interfere with the tested extracts. Their evolution was investigated after 24 h and 48 h, during which the larvae had embryonic energy reserves as lipids [29].

#### 2.6.1. Fluorescent Microscopy

The brine shrimp larvae were stained with 3% acridine orange (Merck Millipore, Burlington, MA, USA) for 5 min. The samples were dried in darkness for 15 min and placed on the microscope slides.

#### 2.6.2. Data Processing

The microscopic images were achieved using a VWR microscope VisiScope 300D (VWR International, Radnor, PA, USA) with a Visicam X3 camera (VWR International, Hilden, Germany) at 40×, 100×, and 400× magnifications and processed with VisiCam Image Analyzer 2.13.

Fluorescent microscopy (FM) images were achieved using an OPTIKA B-350 microscope (Ponteranica, BG, Italy) blue filter (λex = 450–490 nm; λem = 515–520 nm) and green filter (λex = 510–550 nm; λem = 590 nm). These images were obtained at 100×, 200×, and 400× magnification and processed with Optikam Pro 3 Software (OPTIKA S.R.L., Ponteranica, BG, Italy) [30]. All observations were performed in triplicate.

### 2.7. In Vitro Analysis of the Biological Effects of UBE-Loaded Bioadhesive Oral Films on Human Blood Cell Cultures and CLS-354 Tumor Cell Line

#### 2.7.1. Equipment

The study platform for in vitro cytotoxicity analysis was the Attune Acoustic focusing cytometer (Applied Biosystems, Bedford, MA, USA). Before cell analysis, the flow cytometer was first set using fluorescent beads (Attune performance tracking beads, labeling, and detection, Life Technologies, Europe BV, Bleiswijk, Netherlands [31], with standard size (four intensity levels of beads population). The cell quantity was established by countering the cells below 1 µm. Over 10,000 cells per sample were gated by Forward Scatter (FSC) and Side Scatter (SSC) for each analysis.

#### 2.7.2. Data Processing

Flow cytometry results were collected using Attune Cytometric Software v.1.2.5, Applied Biosystems, 2010 (Bedford, MA, USA).

#### 2.7.3. Human Blood Cells Cultures

Blood samples were collected into heparinized vacutainers and used throughout the experiment. The blood (1.0 mL) was introduced in 6.0 mL of Dulbecco’s phosphate buffered saline (with MgCl_2_ and CaCl_2_) medium, supplemented with 10% bovine fetal serum, L-glutamine, and antibiotics mix solution. The mixture was placed in untreated Nunclon Vita Cell culture 6-well plates (Kisker Biotech GmbH & Co.KG, Steinfurt, Germany) were incubated in a Steri-Cycle™ i160 CO_2_ Incubator (Thermo Fisher Scientific Inc., Waltham, MA, USA), with 5% CO_2_, at 37 °C for 72 h. Next, blood cell cultures were treated with the samples and controls and incubated for 24 h in the same conditions. All the flow-cytometry analyses were performed after this incubation time.

#### 2.7.4. CLS-354 Cell Line, Cell Culture

The CLS-354 cells were cultured for 7 days in DMEM High Glucose with 10% FBS supplemented with antibiotic mix solution in a 5% CO_2_ humidified atmosphere at 37 °C, according to https://www.clsgmbh.de/pdf/cls-354.pdf (accessed on 5 March 2022). Then, the cells were dissociated with Trypsin-EDTA and centrifugated at 3000 rpm for 10 min in a Fisher Scientific GT1 Centrifuge (Thermo Fisher Scintific Inc., Waltham, MA, USA). Then, the cells were distributed in Millicell™ 24-Well Cell Culture Microplates (Termo Fisher Scientific Inc., Waltham, MA, USA). After treatment, the cells were incubated for 24 h in the same conditions. All the flow-cytometry analyses were performed after this incubation period.

#### 2.7.5. Samples and Control Solutions

F-UBE-HPC was dispersed in 1 mL of suitable culture media for both types of cells with 1% DMSO and incubated at 37 °C for 24 h. Usnic acid of 125 µg/mL in 1% DMSO was selected as positive control; 1%DMSO was also the negative control.

#### 2.7.6. Annexin V-FITC Apoptosis Assay

The cells were placed in flow-cytometry tubes (with 2 µL Annexin V-FITC and 2 µL PI of 20 µg/mL) and incubated at room temperature, in darkness, for 30 min. Then, 1 mL of FCB was added, and viable cells, early apoptotic cells, late apoptotic cells, and necrotic cells were examined with a flow cytometer using a 488 nm excitation, green emission for Annexin V-FITC (BL1 channel), and orange emission for PI (BL2 channel).

#### 2.7.7. Evaluation of Caspase 3/7 Activity

A total of 300 µL of cell cultures was transferred in flow-cytometry tubes; then, 20 µL of Magic Red^®^ Caspase-3/7 Substrate-MR-(DEVD)_2_-solution was added and well-mixed with the cells. Next, 20 µL of PI was added. After incubation, 1 mL FCB was added. Then, the early stages of cell apoptosis by activating caspases 3/7 (DEVD-ases, according to https://www.abcam.com/caspase-37-assay-kit-magic-red-ab270771.html (accessed on 3 April 2022) were analyzed through flow cytometry using a 488 nm excitation, red emission for MR-(DEVD)_2_-BL3 channel, and orange emission for PI-BL2 channel.

#### 2.7.8. Evaluation of Nuclear Condensation and Lysosomal Activity

A Magic Red^®^ Caspase-3/7 Assay Kit containing Hoechst 33,342 stain (200 μg/mL) and acridine orange (AO, 1.0 µM) was used. Hoechst 33,342 is a cell-permeant nuclear stain; when it is linked to double chain DNA, it emits blue fluorescence, highlighting condensed nuclei in apoptotic cells.

Acridine orange is a chelating dye that can be used to reveal lysosomal activity-https://www.abcam.com/caspase-37-assay-kit-magic-red-ab270771.html (accessed on 3 April 2022). Therefore, 300 µL of cell culture was introduced in flow-cytometry tubes; then, 2 µL of Hoechst 33,342 stain was added, and the cells were well-mixed. Then, 50 µL of acridine orange 1.0 µM was added, and the cells were incubated in darkness, at room temperature, for 30 min. Finally, 1 mL FCB was added, permitting the examination of the cells at the flow cytometer. UV excitation and blue emission for Hoechst 33,342 (VL2) at 488 nm and green emission acridine orange (BL1 channel) were used for examination.

#### 2.7.9. Evaluation of Total ROS Activity

A total of 100 µL of stain solution from ROS Assay was added to each 1 mL of cell culture in flow-cytometry tubes and well-mixed. After incubation at 37 °C, in 5% CO_2_, for 60 min, the cells were examined with a flow cytometer using a 488 nm excitation and green emission for ROS (BL1 channel).

#### 2.7.10. Cell Cycle Analysis

A total of 1 mL of cell culture was washed in FCB and fixed for 10 min with 50 µL ethanol. Next, the cells were treated with PI (20 µg/mL) and RNase A (30 µg/mL) and incubated at room temperature, in darkness, for 30 min. Then, 1 mL FCB was added, and the cell cycle distribution was examined by flow cytometry using a 488 nm excitation and orange emission for PI (BL2 channel).

#### 2.7.11. Evaluation of Cell Proliferation

Volumes of 1 mL of cell cultures were incubated with 50 µM EdU (500 µL) at 37 °C for 2 h. Then, the cells were fixed (with 100 µL 4% paraformaldehyde in PBS) and permeabilized (with 100 µL Triton X-100). After washing in 3% buffer sodium azide (BSA) and centrifuging at 300 rpm for 5 min, at 4 °C, the cells were incubated with a reaction mix (500 µL) for 30 min at room temperature, in darkness. Then, they were washed in permeabilization buffer and centrifuged at 300 rpm for 5 min, at 4 °C. After these procedures, 1 mL FCB was added, and the cells were examined by flow cytometry using a 488 nm excitation and green emission for EdU-iFluor 488 (BL1).

### 2.8. Data Analysis

All analyses were performed in triplicate, and the results were presented as means values ± standard deviation (SD). The results are presented as percent (%) of cell and nuclear apoptosis, caspase 3/7 activity, autophagy, cell cycle, DNA synthesis, and count (×10^4^) of oxidative cellular stress after flow-cytometry analyses were performed with SPSS v. 23 software, IBM, Armonk, NY, USA, 2015. The Levene test was analyzed for homogeneity of variances of samples. At the same time, a paired *t*-test was used to evidence the differences between sample and controls; *p* < 0.05 was established as statistically significant. The Principal Component Analysis was achieved with XLSTAT v. 2022.2.1. by Addinsoft (New York, NY, USA).

## 3. Results

### 3.1. Development of the Bioadhesive Oral Films

Both films are transparent, flexible, and homogenous with smooth surfaces (Appendix A). F-UBE-HPC (Appendix A) has a yellowish-green, faint-brown color, while R is colorless. (Appendix A). The patches peeling and cutting processes occurred without forming air bubbles, cracks, or other defects.

Each UBE-loaded bioadhesive oral film has 0.330 mg UBE, with a total phenolic content (TPC) of 573.234 mg/g (one film has a TPC of 0.189 mg) and 108.742 mg/g usnic acid (one film contains 0.036 mg UA).

### 3.2. Physico-Chemical Characterization of the Bioadhesive Films

#### 3.2.1. SEM Analysis

Figure 1 shows the SEM images of Reference (R) and F-UBE-HPC. A clear difference can be noticed between their aspect. Thus, R has a rough surface (Figure 1a), while F-UBE-HPC shows an almost entirely smooth one (Figure 1b).

#### 3.2.2. AFM Analysis

The AFM 2D images at the scale of (8 × 8) µm^2^ in the so-called “enhanced contrast” view mode are presented in Figure 2a,b.

The first row shows the Reference (R) (Figure 2a). Some differences in the morphology of the parental patch can be noticed as the sample R appears more compact with local accumulations of material (as indicated with a few arrows in Figure 2a). Some large pits (surface cavities) are also observed on the R surface. The arbitrary lines plotted below the AFM images of R suggest a similar z-scale (level oscillations) of about 120 nm (the vertical scale of the plotted line scans in Figure 2a).

The morphology of F-UBE-HPC (Figure 2b) is much more uniform compared with the parental sample, exhibiting small protuberances. There is a decrease in roughness (Rq), as also observed in the line scan from Figure 2b, where a vertical scale of ~50 nm can be observed (from −30 to +20 nm). Moreover, the peak-to-valley (Rpv) parameter decreases after “functionalization” (from R to F-UBE-HPC), as visible in Figure 2c. The histograms of the roughness (Rq) and peak-to-valley (Rpv) (Figure 2c) were plotted for all the line scans, indicated by red horizontal lines in the AFM images. When the corrugation parameters are calculated for the whole scanned area (i.e., (8 × 8) µm^2^), as displayed in Figure 2d, it was observed that the R and F-UBE-HPC samples are roughed.

Relevant images of both films’ morphology are presented in Figure 2e,f at the scale of (3 × 3) µm^2^, choosing some areas as flat as possible. In this case, it is evident that locally, meaning in small areas, the films are relatively uniform regarding the morphological characteristics but maintain the same trend regarding their roughness.

#### 3.2.3. FTIR Analysis

FTIR spectroscopy was used to study the formation of bioadhesive films. The FTIR spectra of both films (R and F-UBE-HPC) are shown in Figure 3.

In Figure 3, for the peak’s optimal visualization, evidencing the differences between both films, the spectral range was divided into two parts: 4000–2000 cm^−1^ (a) and 2000–400 cm^−1^ (b); moreover, R was marked with black color and F-UBE-HPC with blue.

The FTIR spectrum of pure HPC shows absorption bands at 3431 cm^−1^ due to the OH group from the pyranose unit, at 2967 cm^−1^ and 2932 cm^−1^ due to CH2 and CH stretching vibrations, at 1533 cm^−1^ due to C=C stretching vibration, and at 1079 cm^−1^ due to C-O-C stretching vibration [32].

The FTIR spectrum of PEG shows a broad band around 3500 cm^−1^ due to the OH group forming hydrogen bonds with other polymers. The characteristic bands appearing at 1724 cm^−1^ and 1633 cm^−1^ are attributed to the carbonyl group (C=O) and stretching vibrations of CH, CH2, and CH3 groups. The bands at 1464 and 1343 cm^−1^ are associated with the C-H bending. The stretching vibrations of the C-C-O group appear at 1280 and 1100 cm^−1^. At around 950 and 840 cm^−1^, the harmonic bands of the C-C-O group appear [33].

The FTIR spectra of R and F-UBE-HPC bioadhesive films show almost the same peaks as in the pure polymers but with some displacement. This aspect proves that between the two polymers exists a strong interaction. The UBE presence reduces the FTIR peak intensity compared with R (Figure 3—blue line), evidencing its complete dispersion in the polymer matrix.

#### 3.2.4. XRD Analysis

Figure 4a shows the XRD patterns of the reference R and F-UBE-HPC films. According to Ishii et al. [34], the XRD pattern of HPC showed an amorphous characteristic, with a broad peak at 2θ° = 9° and 20°.

In the XRD pattern of R, the amorphous character was observed with the two broader peaks characteristic of the HPC compound. The amorphous aspect is maintained when UBE is loaded, but the peak intensities are reduced, thus proving that UBE was successfully dispersed in the polymer matrix.

#### 3.2.5. TG Analysis

Thermogravimetric and differential thermal analyses were performed to assess the bioadhesive films’ thermal behavior and stability. Reference (R) and F-UBE-HPC exhibited similar behavior when a heating program from 25 to 600 °C (Figure 4b) was performed.

In the first step (between 25 and 100 °C), the films’ mass loss (of 2.4% for R and 2.3% for F-UBE-HPC) can be associated with the loss of residual solvent and physisorbed water. The decomposition process of the organic compounds occurs in two distinct steps, between 190–380 °C and 380–550 °C. The mass losses and maximum decomposition temperatures obtained from DTA curves associated with the three steps are presented in Table 2.

### 3.3. Pharmacotechnical Properties of the Bioadhesive Oral Films

The pharmacotechnical properties evaluated in this study are displayed in Table 3. Data from Table 3 show that the F-UBE-HPC average weight reported no significant difference compared with R (110 ± 4.77 vs. 107 ± 5.25, *p* > 0.05). Both films have a thickness of 0.093 mm, with minimal differences in standard deviation (SD) values between F-UBE-HPC and R (0.003 vs. 0.002).

Both developed films displayed a high folding endurance, above 300. Moreover, tensile strength and elongation did not show significant differences between F-UBE-HPC and R (2.48 ± 1.58 vs. 2.57 ± 1.71 and 63.14 ± 1.94 vs. 62.75 ± 1.52, *p* > 0.05). F-UBE-HPC’s moisture content was similar to the R one (6.58 ± 0.44 vs. 6.23 ± 0.56, *p* > 0.05). Both bioadhesive films have a neutral pH, without noticeable differences between F-UBE-HPC and R (7.10 ± 0.02 vs. 7.07 ± 0.01, *p* > 0.05). The films displayed a fast disintegration time of 118 ± 3.16 (F-UBE-HPC) vs. 115 ± 4.19 (R).

The swelling rate over the 6 h of study is presented in Figure 5.

Table 3 and Figure 5 show that F-UBE-HPC and R have the same swelling performance after 6 h (289 ± 5.82 vs. 288 ± 6.13, *p* > 0.05).

Regarding the ex vivo bioadhesion time, both oral films (F-UBE-HPC and R) show similar values: 98 ± 3.58 vs. 98 ± 4.17, *p* > 0.05 (Table 3).

### 3.4. Antimicrobial Activity

The standard drugs and F-UBE-HPC microdilutions used in antimicrobial activity evaluation are registered in Table 4.

F-UBE-HPC inhibitory activity on both bacterial and fungal strains was dose-dependent, and the obtained data are displayed in Table 5 and Figure 6.

Table 5 shows that F-UBE-HPC of [5.5–0.085] mg/mL concentration act on *S. aureus* similar to CTR of [1.5–0.024] mg/mL (Table 5). On *P. aeruginosa,* their inhibitory activity is higher: in the range of [5.5–0.685] mg/mL and act similarly to CTR of [2.498–0.187] mg/mL. The final F-UBE-HPC concentration range of [0.342–0.085] mg/mL inhibits bacterial colonies proliferation such as a CTR of [0.093–0.024] mg/mL.

Figure 6 shows that F-UBE-HPC of [5.5–2.75] mg/mL induces *C. albicans* partial death and a low proliferation of *C. parapsilosis.* At the following concentration range [1.375–0.171] mg/mL, it similarly acts on both *Candida* sp., determining a moderate proliferation. The final concentration (0.085 mg/mL) leads to *C. albicans* fast proliferation and *C. parapsilosis* moderate.

### 3.5. Cytotoxic Activity on A. salina Larvae

After 24 h, the larvae were alive, swimming, and showing normally visible movements. We investigated them under a microscope to observe the changes after 24 and 48 h of exposure. All these microscopic images are presented in Figure 7 and Figure 8.

After 24 h of exposure, *A. salina* larvae showed an early digestive blockage, with contraction of the digestive tube, a low amount of food in the upper digestive part, and a low detachment of cuticle from peripheral tissues (Figure 7e–h). After 48 h, 35.82% of larvae were alive, and 8.95% were in a sublethal stage. The registered mortality was 55.22% due to massive digestive blockage and tissue destruction (Figure 7m–p).

The FM images show *A. salina* larvae after 48 h exposure at F-UBE-HPC (Figure 8d–f) compared with blank (Figure 8a–c). Figure 8e,f shows intracellular lysosomes activated in the cell death process, evidenced by red fluorescence.

### 3.6. In Vitro Analysis of the Biological Effects of UBE-Loaded Mucoadhesive Oral Films on Human Blood Cell Cultures and CLS-354 Cancer Cell Line

The effects of F-UBE-HPC on blood cell cultures and CLS-354 tumor cell line were studied by highlighting the biological mechanisms implied in caspases 3/7 activity, nuclear shrinkage, autophagy, oxidative stress, cell cycle, apoptosis, and DNA synthesis and fragmentation, following previous preliminary studies with *U. barbata* dry ethanol extract [35,36].

#### 3.6.1. Caspase 3/7 Activity

To prove that F-UBE-HPC causes cell apoptosis in normal blood cells and OSCC tumor cells, the effector caspase 3/7 intracellular activity was determined by flow cytometry. The results are illustrated in Figure 9.

After 24 h of treatment, the caspase-3/7 activation mechanisms present significantly more decreased values than the negative and positive controls (13.56 ± 3.13 vs. C1: 29.26 ± 1.97, *p* < 0.05; C2UA: 44.74 ± 0.41, *p* < 0.01, Figure 9A–C,G).

The biochemical cascade of reactions implied in the proapoptotic signal after F-UBE-HPC treatment in the CLS-354 tumor cell line registers a considerable augmentation reported to 1% DMSO negative control and 125 µg/mL UA positive control (53.42 ± 3.16; vs. 21.88 ± 5.09; 27.02 ± 1.64, *p* < 0.01, Figure 9D–F,H).

#### 3.6.2. Nuclear Condensation and Lysosomal Activity

After 24 h of F-UBE-HPC treatment in normal blood cells and CLS-354 tumor cells, the presence of pyknotic nuclei (stained with Hoechst 33342) and the lysosomal activity (evidenced by acridine orange) are displayed in Figure 10.

Nuclear shrinkage, after 24 h treatment with F-UBE-HPC in normal blood cell cultures had significantly lower values (4.23 ± 0.58) than 1% DMSO negative control: 24.50 ± 2.21, *p* < 0.01 (Figure 10A,B,M), and appreciably higher than 125 µg/mL UA positive control: 3.19 ± 0.30, *p* < 0.05 (Figure 10A,C,M). The autophagy levels strongly decreased compared with both controls: 5.36 ± 0.65; vs. C1: 51.30 ± 3.25; C2UA: 27.05 ± 1.52, *p* < 0.01 (Figure 10G–I,M).

Flow-cytometry examination by Hoechst 33342/acridine orange dual stain of tumor cells after F-UBE-HPC treatment showed chromatin condensation (NS) and autophagy (A) significant elevation compared with 1% DMSO negative control: NS: 35.36 ± 2.28 vs. 16.11 ± 3.11; A: 71.81 ± 1.47 vs. 12.57 ± 0.92, *p* < 0.01; Figure 10D,E,J,K,M).

As shown in Figure 10D,F,J,L,N, F-UBE-HPC displays significantly lower values of nuclear shrinkage (NS) and appreciably higher autophagy (A) levels reported to positive control with 125 µg/mL UA (NS: 35.36 ± 2.28 vs. 44.03 ± 0.36; A: 71.81 ± 1.47 vs. 53.35 ± 2.63, *p* < 0.05).

#### 3.6.3. ROS Levels

In blood cell cultures and CLS-354 tumor cell line, F-UBE-HPC induced powerful oxidative stress expressed as total ROS level measured through flow cytometry. The obtained results are indicated in Figure 11.

ROS levels substantially increased in the blood cells treated with F-UBE-HPC compared with the 1% DMSO negative control and positive control of 125 µg/mL of UA (2833.33 × 104 ± 152.75; vs. 242.00 × 104 ± 2.00; 846.66 × 104 ± 5.77; *p* < 0.01, Figure 11A–C,G,I).

Oxidative stress was strongly augmented in the OSCC cell line after 24 h contact with F-UBE-HPC compared with the 1% DMSO negative control and 125 µg/mL of UA positive control: 3100.00 × 104 ± 100.00; vs. 15.66 × 104 ± 4.04; 966.66 × 104 ± 57.73, *p* < 0.01 (Figure 11D–F,H,J).

#### 3.6.4. Cell Cycle Analysis

Human blood consists of leukocytes (white blood cells, WBC), thrombocytes (platelets), and erythrocytes (red blood cells, RBW). Only leukocytes contain a cell nucleus, essential when estimating the amount of DNA in human blood cell cultures [37].

DNA content was performed by propidium iodide/RNase stain to explore the effects of F-UBE-HPC in normal blood cells and CLS-354 tumor cells (Figure 12).

As shown in Figure 12A,B,G,I, in normal blood cell cultures, F-UBE-HPC induces a higher cell cycle arrest in G1/G0 phase (93.91 ± 1.41) than the negative control (88.52 ± 0.54, *p* < 0.05). DNA synthesis phase of the cell cycle presented significantly more decreased values than both controls: 0.99 ± 0.24 vs. C1: 4.76 ± 0.68, C2UA: 2.86 ± 0.23; *p* < 0.01 (Figure 12A–C,G,I).

CLS-354 tumor cells were treated for 24 h with F-UBE-HPC to understand whether cell growth inhibition was due to cell cycle arrest. The results show a significant G0/G1 phase increase compared with the 1%DMSO negative control (94.66 ± 1.01 vs. 92.13 ± 1.61, *p* < 0.05 (Figure 12D,E,H,J). DNA synthesis reported considerably lower values reported to 1% DMSO negative control: 0.73 ± 0.43 vs. 5.47 ± 0.83, *p* < 0.01 (Figure 12D,E,H,J).

#### 3.6.5. Apoptosis

The effects of F-UBE-HPC treatment in normal blood cells and CLS-354 tumor cells were achieved by examining morphology and cell membrane integrity with annexin V-FITC/PI stain (Figure 13).

In normal blood cells, after 24 h, F-UBE-HPC did not induce early cell apoptosis (0.00 ± 0.00), reporting significant differences compared with the positive control (C2UA: 37.04 ± 0.66, *p* < 0.01). Consequently, cell viability shows appreciably elevated values: 98.35 ± 1.87 vs. 61.43 ± 0.88, *p* < 0.01, Figure 13A,C,G).

In the CLS-354 tumor cell line, similar results were obtained after treatment for 24 h with F-UBE-HPC compared with C2UA (EA: 0.00 ± 0.00 vs. 12.92 ± 1.35, *p* < 0.01 and V: 99.79 ± 1.12 vs. 54.05 ± 1.68, *p* < 0.01) as shown in Figure 13D,F,H).

#### 3.6.6. Cell Proliferation

DNA synthesis (S) and fragmentation (subG0/G1) by EdU incorporation in both cell types were examined to observe the effects of 24 h treatment with F-UBE-HPC (Figure 14).

In normal blood cell cultures, F-UBE-HPC determined considerably lower levels of DNA synthesis than the negative and positive controls (2.44 ± 0.40 vs. C1:10.36 ± 1.21; *p* < 0.01; C2UA: 6.49 ± 1.25, *p* < 0.05). However, DNA fragmentation (implied in apoptosis) significantly increased compared with the 125 µg/mL UA positive control (3.13 ± 1.03 vs. 0.00 ± 0.00, *p* < 0.05, Figure 14A–C,G,I).

On the other hand, in CLS-354 tumor cells, F-UBE-HPC blocked DNA synthesis: (0.00 ± 0.00 vs. C1: 12.44 ± 2.80, and C2UA: 3.14 ± 0.50, *p* < 0.05), while the apoptosis represented by fractional DNA (subG0/G1 phase) decreased compared with the 1% DMSO negative control: 3.19 ± 1.86 vs. 15.18 ± 2.17, *p* < 0.05 (Figure 14D–F,H,J).

#### 3.6.7. Principal Component Analysis

The Principal Component Analysis (PCA) was performed for F-UBE-HPMC oral mucoadhesive patches and both controls (C1-DMSO and C2UA) and variable parameters determined in both cell types (normal blood cells and CLS-354 tumor cells) according to the correlation matrix and PCA-Correlation circle from the Appendix A. The results are displayed in Figure 15.

The two principal components explained the total data variance, with 51.43% attributed to the first (PC1) and 48.57% to the second (PC2). The PC1 was associated with both controls (C1-DMSO and C2UA), viability, early and late apoptosis, nuclear condensation, and DNA fragmentation (subG0/G1) in both cell types and cell cycle arrest in G0/G1 and necrosis in CLS-354 tumor cells. The PC2 was related to F-UBE-HPC, ROS, and caspase 3/7 activity, DNA synthesis and autophagy in both cell types, and cell cycle arrest in G0/G1 and necrosis in normal blood cells.

In normal blood cells, caspase 3/7 activity has a high positive correlation with early and late apoptosis (*r* = 0.864, *p* > 0.05) and a low one with DNA synthesis and autophagy (*r* = 0.514 and 0.475, *p* > 0.05). It moderate negatively correlates with ROS (*r* = −0.735, *p* > 0.05) and cell cycle arrest in G0/G1 (*r* = −0.698, *p* > 0.05). Cellular oxidative stress (ROS) shows a strong positive correlation with subG0/G1 (*r* = 0.999, *p* < 0.05). It highly negatively correlates with DNA synthesis and autophagy (*r* = −0.959 and 0.946, *p* > 0.05) and moderately with necrosis and nuclear condensation (*r* = −0.796 and 0.648, *p* > 0.05).

In CLS-354 tumor cells, ROS levels are considerably correlated with caspase 3/7 activation (*r* = 0.988, *p* > 0.05) and autophagy (*r* = 0.908, *p* > 0.05). Caspase 3/7 activity also highly correlates with autophagy, but the *r* value is slightly lower compared to ROS (*r* = 0.834, *p* > 0.05). Oxidative stress reports a high negative correlation with DNA synthesis (*r* = −0.880, *p* > 0.05) and a moderate one with subG0/G1 (*r* = −0.675, *p* > 0.05). Caspase 3/7 activity is also negatively correlated with both processes, but the *r* values are lower than ROS ones (*r* = −0.797 and −0.554, *p* > 0.05). Moreover, caspase 3/7 activity moderately correlates with cell cycle arrest in G0/G1 (*r* = 0.621, *p* > 0.05).

By correlating and interpreting these data, the places of F-UBE-HPC patches and both controls (C1-DMSO and C2UA) in the PCA-correlation biplot (Figure 15) were explained, highlighting the corresponding processes triggered in CLS-354 cancer cells and normal blood cells and their complementary mechanisms.

## 4. Discussion

Until the preparation of UBE-loaded bioadhesive oral films analyzed in the present work for potential applications in oral cancer treatment, *U. barbata* lichen was thoroughly studied for almost six years.

First, it is essential to mention that this lichen was harvested from a coniferous forest in the same area (47°28′ N, 25°12′ E, and 900 m altitude), belonging to the highest Romanian volcanic mountains (Calimani mountains) [30,38]. Coniferous forest soil is adjacent to the Tinovul Mare Poiana Stampei peat bog with a natural origin, its accumulation beginning in the post-glacial period [39]. The specific conditions of the *U. barbata* native zone consist of seasonal water-level fluctuations with thermic variations between −1 °C and 14 °C, with a precipitation range of 600–800 mL [39]. The soil has pH values between 4.09 and 5.89 and contains several trace/heavy metals under the alert threshold from national protocols [39].

A detailed elemental analysis was performed on dried lichen harvested in 2020 [30,38]. Twenty-three metals were investigated by inductively coupled plasma mass spectrometry (ICP-MS) [38]. The heavy metal concentrations were associated with those reported by Cazacu et al. [39] in the forest soil, and the results reveal that autochthonous *U. barbata* contains all, in lower amounts than the permissible limits for medicinal plants. However, the mercury content (0.671 ± 0.020 µg/g) was lower than the WHO’s and FDA’s approvable limit (1 µg/g) and over the one mentioned in the European Pharmacopoeia (0.1 µg/g) [38]. The metal concentrations were compared with those reported in the scientific literature regarding *U. barbata* from unpolluted zones (Mountain Natural Park in Bulgaria [40], Sri Lanka rain forest [41] and central and southern Tierra del Fuego, Patagonia, Argentina [42]). Thus, the data obtained confirmed the autochthonous lichen’s suitability for pharmaceutical use [30], according to regulatory agencies [43], which restrict the harvesting area for accurate results in human-safe formulations.

The loss of drying of the dried lichen was determined [44], and bioactive secondary metabolites were identified and quantified [45]. The research on *U. barbata* from the Calimani mountains continued with the preparation of various extracts in different solvents through low-cost and easy-to-use conventional processes [38,44,46], investigating the suitable solvents and extraction conditions to obtain *U. barbata* extracts with pharmacological potential. All extracts were comparatively analyzed, quantifying their metabolites through HPLC-DAD [45,47] and evaluating their antioxidant, antimicrobial, and cytotoxic potential. The cytotoxicity prescreen was performed using *A. salina* as an animal model [44], continuing with complex studies on normal and tumor cells [35,36].

This previously described interdisciplinary research on *U. barbata* from the Calimani mountains led to selecting the most appropriate extracts for bioadhesive oral film formulation, including the dry ethanol extract. UBE contains usnic acid and other various phenolic compounds [48,49,50], including common polyphenols (ellagic acid and gallic acid) with high pharmacological potential [51]. Thus, *U. barbata* ethanol extract shows a dose-dependent inhibitory activity on the most known bacteria responsible for oral infectious disease [52] in immunocompromised patients: *S. aureus, S. pneumoniae, S. pyogenes, Enterococcus* sp., other Gram-positive bacteria isolated from oral cavity and pharynx *(S. epidermidis, S. oralis, S. intermedius)*, and *P. aeruginosa* [38,51,53,54]. It also reported inhibitory effects on *C. albicans* [38]. Phenolic compounds extracted in ethanol have antibacterial and antifungal properties and could act synergistically in UBE. Moreover, inhibitory activity on bacterial strains varies in direct proportion to antioxidant potential [55], and UBE exhibits an intense antioxidant activity [51]. In normal cells, it shows antioxidant and protective effects, neutralizing a part of the free radicals generated by 0.2%DMSO, thus reducing the lipids peroxidation [56] and activity of antioxidant enzymes Superoxide dismutase (SOD) and Catalase (CAT) induced by cellular oxidative stress [36]. Contrarily, UBE demonstrated an in vitro anticancer effect on CAL-27 tumor cell line, triggering cell death processes. It acts as a prooxidant, increasing the oxidative stress induced by 0.2% DMSO and thus stimulating cellular antioxidant defense—Glutathione peroxidase (GPx) and SOD—and augmenting the malondialdehyde (MDA) levels [36,57].

Knowing the UBE bioactive metabolites’ content and its pharmacological potential, the present work aimed to study a pharmaceutical formulation containing this *U. barbata* dry extract. Thus, UBE-loaded bioadhesive oral films were prepared and analyzed, their properties being compared with References (films without UBE). As a film-forming polymer, HPC was selected due to its non-ionic character and water solubility. In addition, it proved to be physiologically inert, non-irritating, bioadhesive, and biodegradable [58,59,60,61]. PEG 400 was used as a plasticizer in a 5% (*w*/*w*) amount to improve the film flexibility and obtain suitable mechanical properties [62,63]. Both films are transparent, flexible, and homogenous. The yellowish-green, faint-brown color of F-UBE-HPC is due to lichen dry ethanol extract, and its morphology shows a smooth surface.

The physical and chemical investigation of the bioadhesive oral films containing *Usnea barbata* dry ethanol extract demonstrated the successful incorporation of the UBE in the polymer matrix. The pharmacotechnical analysis of F-UBE-HPC was realized according to other previously published studies on bioadhesive films containing plant extracts [64,65].

Therefore, both films’ thickness values show that UBE does not influence their dimensions. Because the films’ thickness is directly connected to active ingredient concentration and the polymer’s mucoadhesive function, uniformity is required. The selected plasticizer type and amount also influence the film thickness. Generally, the suitable thickness for bioadhesive oral films is 0.05–1.0 mm [66,67,68], and the UBE-loaded formulation belongs to this. The film’s considerable folding endurance highlights appreciable flexibility and mechanical resistance provided by PEG 400 (used as a plasticizer in a 5% amount). The plasticizer’s role is to modify the viscoelastic properties of the polymer, which will subsequently influence the adhesion ability and release behavior. Moreover, the UBE load did not change the films’ mechanical properties.

The bioadhesive films’ elongation evidenced their remarkable flexibility. The tensile strength indicates the films’ high resistance, suitable to withstand the mechanical stress that occurs during unpacking and appliance on the oral mucosa. The films’ hardness depends on the type and amount of polymer and plasticizer used. The obtained results prove the accurate selection of the excipients appropriate to incorporate UBE as an active ingredient in achieving high-quality bioadhesive films. Moreover, the films’ flexibility is essential in aiding the penetration of active ingredients and attachment to the oral mucosa. Bharkatiya et al. [69] stated that low tensile strength and elongation characterize a soft and weak polymer. In contrast, moderate tensile strength and low elongation are typical for a hard and brittle polymer. Furthermore, high tensile strength and elongation describe a soft and tough polymer, as in the studied formulations.

The flexibility of the chains diminishes as the crosslinking density in water-soluble polymers increases, thus protecting against overhydration. The bound water can build H-links with the HPC’s accessible chains, allowing them to move more freely [70,71]. Repka et al. [72] stated that HPC has less water affinity. The obtained values also indicate a certain amount of moisture content in the films’ structure, but it is well-known that humidity is needed to ensure bioadhesion. Moreover, bioadhesion is improved by functional groups that can create hydrogen bonds.

Both films’ neutral pH ensures their tolerability and biocompatibility with oral mucosa. This property confirms that UBE inclusion in the polymer matrix does not affect the base films’ pharmacotechnical properties; they exclusively depend on the selected excipients. In addition, pH value can influence the polymer’s bioadhesive properties, and a neutral value is optimal for mucoadhesion [73,74]. HPC used as polymer film-forming is responsible for rapid disintegration, and the plasticizer also significantly influences this property. Another essential factor is the manner of active ingredient dispersion in the matrix base. In the studied formulation, UBE was solubilized and mixed with the polymer matrix, thus diminishing the disintegration time.

The swelling index plays an essential role in regulating the release rate of the active ingredient. The diffusion of water molecules into the hydrated decreases the number of hydrogen bonds and increases the strength between the polymers. Panomsuk et al. [75] proved that a polymer chain with a low ability to make hydrogen bonds cannot build a strong matrix structure, and water penetration would be difficult. The polymeric matrix swelling ability depends on its resistance to the migration of the water molecules, and the hydroxyl group is essential for hydrophilic cellulose polymers’ matrix integrity [37]. Moreover, the polymer’s substituted groups significantly limit the active ingredients’ influence on the swelling properties. Hence, the similar swelling ratio of F-UBE-HPC and R demonstrates the impact of the active ingredient dispersion on the swelling degree because UBE incorporated as a solution in F-UBE-HPC did not change the polymer properties.

The swelling behavior also has an impact on the film’s bioadhesive performance. The increased residence time is the result of the elastic behavior induced by the used plasticizer. The interaction between free polymer chains and mucin is essential; the active ingredients loaded into the matrix could occupy the chains [76], proving that their dispersion directly impacts the adhesion performance. The obtained data show that both oral films have the same bioadhesion time, demonstrating that UBE load did not influence the matrix adhesivity.

The F-UBE-HPC rapid disintegration and neutral pH were essential for accurately evaluating their pharmacological potential because the afferent studies were conducted in vitro (using bacterial, blood, and tumor cell cultures) and in vivo (on *A. salina* larvae used as an animal model) [77]. These previously mentioned analyses were performed with homogenous dispersions of UBE-loaded films in different culture media and, respectively, in phosphate buffer. On the other hand, the tested cultured cells and *A. salina* larvae are strongly affected by pH variation; thus, the films’ neutral pH was requested to obtain accurate results.

For optimal evaluation, the studies used for the UBE-loaded films had a similar trend to previous ones for UBE investigation. Thus, the F-UBE-HPC antimicrobial effects were examined on similar bacterial and fungal species, and a cytotoxicity prescreen was performed on *A. salina* larvae. Anticancer activity was evaluated on a similar OSCC cell line (CLS-354, mouth epithelial squamous cell carcinoma). As normal cells, blood cell cultures were used from a single donor (the same as previously described in UBE’s biological studies) because buccal mucosa has a substantial blood supply and is permeable for most blood cells. Therefore, the most common types of WBC in blood—lymphocytes (with T cells) and segmented cells (polymorphonuclear, including neutrophils and eosinophils)—are the predominant leukocytes mixed with various endothelial cell types in the oral mucosa. No significant differences in the proportions of these cells between healthy children and adults indicate that both innate and adaptive immune systems function in the oral cavity [78]. All selected studies reproduced, in a certain measure, the conditions of the previous UBE pharmacological evaluation, aiming to know whether the UBE-loaded films’ activities are similar to the UBE ones.

The results show that F-UBE-HPC displayed dose-dependent inhibitory effects against *S. aureus, P. aeruginosa,* and both *Candida* sp. through various mechanisms belonging to UBE’s phenolic secondary metabolites: inhibiting protein, DNA, and cell wall synthesis, permeabilizing the cell wall and chelating various microbial micronutrients and reducing their disponibility for cell growth [79]. These microbial species are frequently implied in immunocompromised patients’ oral cavity infections. UBE-loaded bioadhesive films could protect them, facilitate their recovery after radiotherapy, and generally increase their life quality. Moreover, the significant inhibitory action against *C. albicans* could also be helpful in oral cancer prevention [80,81].

The BSL assay was used as cytotoxicity prescreen because it is a low-cost, fast, and effective analysis due to *A. salina* larvae sensitivity. It can be adapted to various applications [82,83,84]. The brine shrimp larvae used as biotester have a rapid growth rate and, in 24 h, pass to another developing stage (i.e., nauplii). Their mortality could be associated with blocking DNA synthesis or cell cycle arrest [29]. The movements of larvae and different morphological changes (visualized due to the larvae’s transparency) could be associated with various cytotoxic mechanisms [28,82,85]. Previous studies consider that BSL assay results could predict the antitumor activity of tested products [86,87]. Thus, morphological changes induced by F-UBE-HPC on *A. salina* larvae and intracellular lysosomes activated in cell death processes (evidenced in FM images) anticipate the previously described in vitro anticancer effects on oral epithelial squamous cell carcinoma [86]. The brine shrimp larvae mortality could be associated with the blockage of DNA synthesis and cell cycle arrest in G0/G1 and intracellular lysosomes from FM images with autophagy.

All previously mentioned factors implied in oral cancer development produce oxidative stress (an imbalance between ROS generation and antioxidant defense, leading to high intracellular ROS levels) [88]. Excessive ROS causes oxidative damage to cellular constituents (lipids, proteins, and DNA), hardly affecting normal cells and then contributing to carcinogenesis [89,90]. ROS are also essential in cell signaling, modulating the most significant cell death pathways (mediated by mitochondria, endoplasmic reticulum, and death receptors) [91]. ROS in high concentration could induce caspase 3/7 activation, triggering cellular processes associated with cell death (DNA damage with cell cycle arrest in different phases [92], nuclear condensation [93], blockage of DNA synthesis, and autophagy [88]).

Generally, cancer cells (due to mitochondrial abnormalities) have higher ROS levels compared with normal cells [94]. Therefore, an overproduction of ROS in cancer cells may offer a cancer-specific therapy [95]. F-UBE-HPC acts on CLS-354 cancer cells, considerably increasing cellular oxidative stress and caspase 3/7 activity, triggering nuclear condensation and autophagy, inducing cell cycle arrest in G0/G1, and blocking DNA synthesis. Compared with UA 125 µg/mL, F-UBE-HPC generates an ROS level 3 × higher; the caspase activity level is double that of UA. Moreover, DNA synthesis is completely stopped, and lysosomal activity is substantially increased compared with UA. Even though CLS-354 tumor cell viability is slowly diminished after 24 h, all intracellular and molecular processes leading to tumor cell death are triggered at a significant level.

On the other hand, in normal blood cells, the basal ROS level is lower than the one in the cancer cell line. A moderate ROS level is essential to promote cell proliferation and survival. In blood cell cultures, oxidative stress induced by UBE-loaded films is lower than in the CLS-354 tumor cell line; their viability after 24 h is over 98%. Compared with cancer cells, it could be considered that F-UBE-HPC displays a protective role in blood cell cultures, remarkably diminishing caspase 3/7 activity, nuclear condensation, and lysosomal activity triggered by 1% DMSO.

If any concerns regarding the F-UBE-HPC safety for human health [96] in relationship with usnic acid hepatotoxicity still exist, the following data could be significant. LipoKinetix (Syntrax, Cape Girardeau, MO), the fat-burning diet supplement [97], was a multi-ingredient product, including 100 mg usnic acid per capsule, together with 25 mg of norephedrine hydrochloride, 100 μg of 3,5-diiodothyronine, 3 mg of yohimbine hydrochloride, and 100 mg of caffeine [98]. The recommended daily dose associated with severe liver failure was 3–6 capsules, corresponding to 300–600 mg of usnic acid [99]. One UBE-loaded bioadhesive oral film contains 0.330 mg UBE, corresponding to 0.036 mg UA; hence, over 2700 films should be necessary to ensure the usnic acid content of 100 mg from a single capsule of LipoKinetix.

## 5. Conclusions

In the present study, bioadhesive oral films loaded with *U. barbata* dry ethanol extract were formulated and manufactured using HPC as a polymer matrix and PEG 400 as a plasticizer.

F-UBE-HPCs and References (the same formulation without UBE) were investigated through physicochemical and pharmacotechnical analyses, and the obtained data reported no significant differences, proving the active ingredient’s consistent incorporation.

The F-UBE-HPC properties are optimal for accurately maintaining the lichen extract’s pharmacological potential. The results prove it, recommending UBE-loaded bioadhesive oral films for possible application in oral cancer treatment and prevention. Additional research will optimize the bioadhesive oral film formulation, performing drug release in vivo and clinical studies, aiming to complete this interdisciplinary work and confirm their therapeutical benefits.

## Figures and Tables

**Figure 1 pharmaceutics-14-01808-f001:**
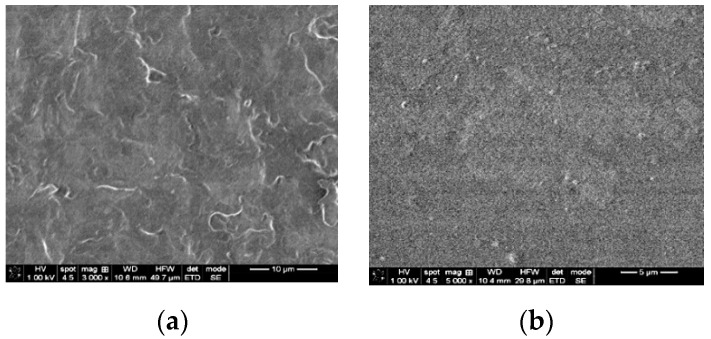
SEM images of (**a**) Reference (R); (**b**) F-UBE-HPC; R—bioadhesive oral film without active lichen extract; F-UBE-HPC—bioadhesive oral film loaded with *U. barbata* dry ethanol extract.

**Figure 2 pharmaceutics-14-01808-f002:**
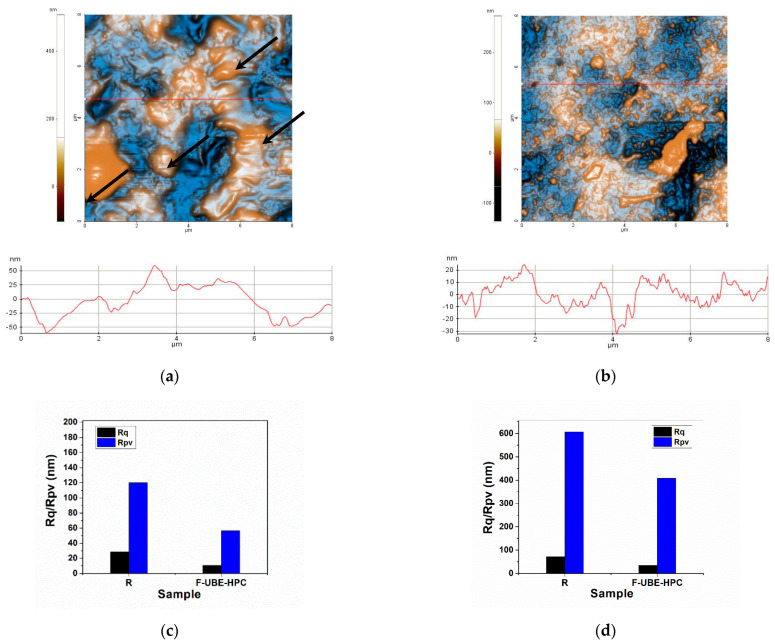
The 2D AFM images (enhanced contrast view) at the scale of (8 × 8) µm^2^ together with representative line-scans for bioadhesive films: (**a**) R, (**b**) F-UBE-HPC; roughness (Rq) and peak-to-valley (Rpv) parameters (**c**) along the line scans and (**d**) for the whole scanned areas; 2D AFM images (enhanced contrast view) at the scale of (3 × 3) µm^2^ together with representative line-scans for the films (**e**) R and (**f**) F-UBE-HPC; R—bioadhesive oral film without active lichen extract; F-UBE-HPC—bioadhesive oral film loaded with *U. barbata* dry ethanol extract.

**Figure 3 pharmaceutics-14-01808-f003:**
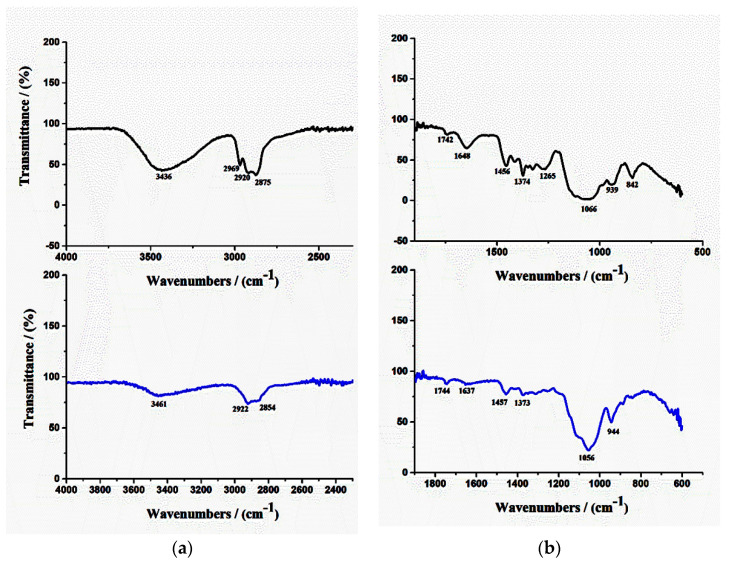
FTIR spectra of R (black line) and F-UBE-HPC (blue line), in the range of 4000–2000 cm ^−1^ (**a**) and 2000–400 cm^−1^ (**b**). R—bioadhesive oral film without active lichen extract; F-UBE-HPC—bioadhesive oral film loaded with *U. barbata* dry ethanol extract.

**Figure 4 pharmaceutics-14-01808-f004:**
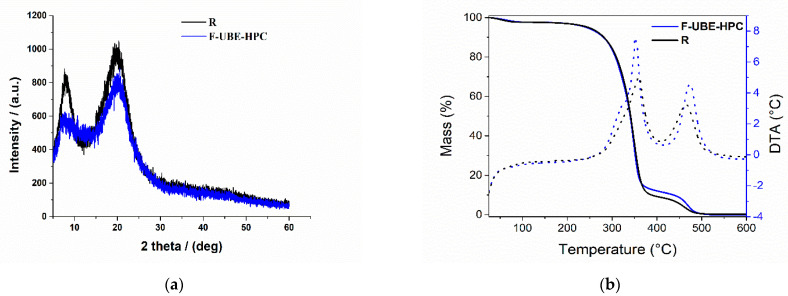
(**a**) XRD pattern and (**b**) TG/DTA curves of bioadhesive oral films: R (black line) and F-UBE-HPC (blue line). R—bioadhesive oral film without active lichen extract; F-UBE-HPC—bioadhesive oral film loaded with *U. barbata* dry ethanol extract.

**Figure 5 pharmaceutics-14-01808-f005:**
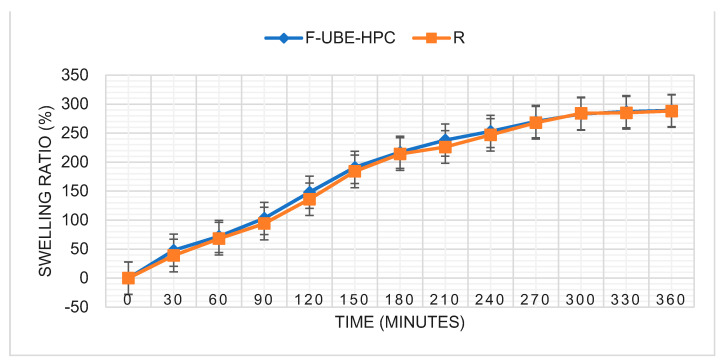
Swelling rate over 6 h for F-UBE-HPC and R; F-UBE-HPC—bioadhesive oral film loaded with *U. barbata* dry ethanol extract; R—bioadhesive oral film without active lichen extract.

**Figure 6 pharmaceutics-14-01808-f006:**
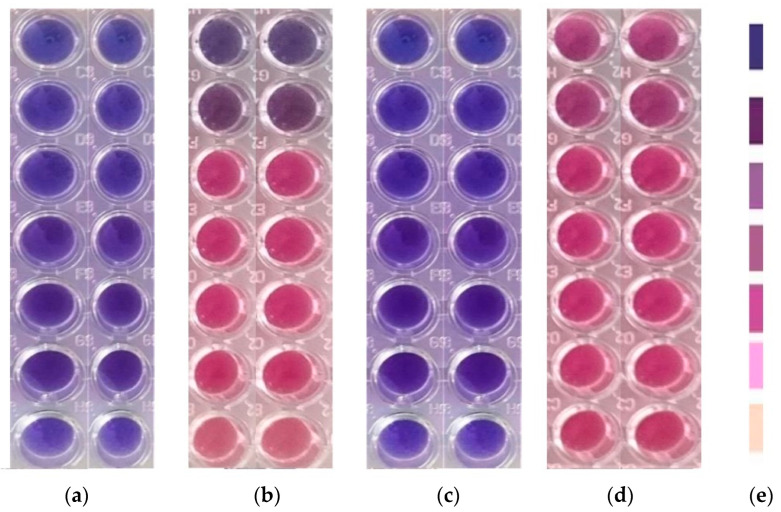
Inhibitory activity of F-UBE-HPC (**b**,**d**) on *C. albicans* (**a**,**b**) and *C. parapsilosis* (**c**,**d**). The results were compared with Terbinafine (**a**,**c**) as an antifungal drug. (**e**) Results interpretation adapted from Bitacura et al. [27], as follows: blue—cells are dead; violet–blue—cells are partially dead; violet—cells are alive, no proliferation; light-violet—low proliferation; dark pink—moderate proliferation; pink—fast proliferation; light pink—very fast proliferation. F-UBE-HPC—bioadhesive oral film loaded with *U. barbata* dry ethanol extract.

**Figure 7 pharmaceutics-14-01808-f007:**
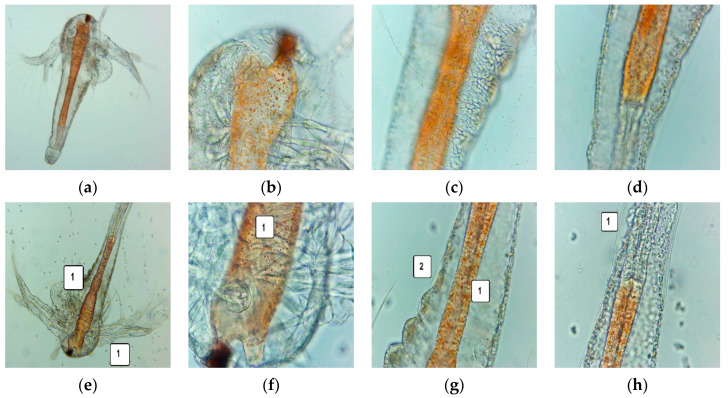
*A. salina* larvae after 24 and 48 h exposure to F-UBE-HPC—microscopic images at 100× (**a**,**e**,**i**,**m**) and 400× (**b**–**d**,**f**–**h**,**j**–**l**,**n**–**p**). After 24 h: blank (**a**–**d**) and sample (**e**–**h**); after 48 h: blank (**i**–**l**) and sample (**m**–**p**). (**e**) contraction of the digestive tube in the upper and middle portion (1); (**f**) reduced amount of food in the upper digestive tube (1); (**g**) early digestive blockage (1), low detachment of the cuticle from larval tissue (2); (**h**) low detachment of cuticle from the peripheral larval tissues (1); (**m**) dead larvae with massive digestive blockage (1) and tissue destruction (2); (**n**) digestive blockage (1) and massive tissue damage (2,3); (**o**,**p**) low quantity of foods in the digestive tube (1), massive tissue destruction (2) and high detachment of the cuticle from larval tissues (3); F-UBE-HPC—bioadhesive oral film loaded with *U. barbata* dry ethanol extract.

**Figure 8 pharmaceutics-14-01808-f008:**
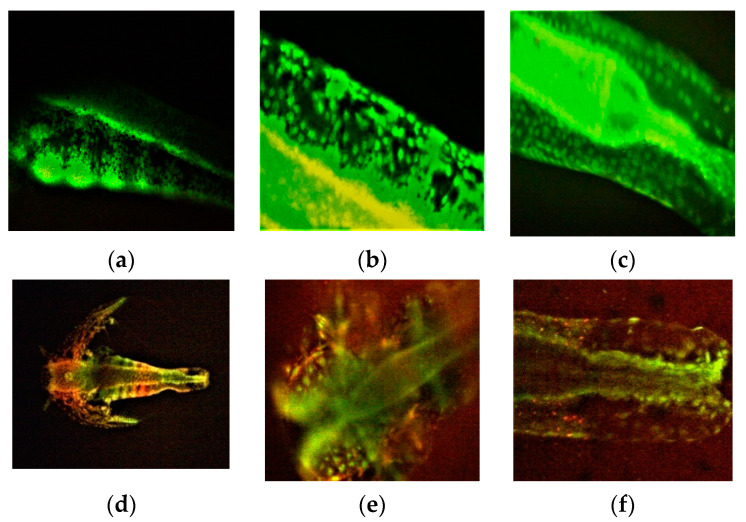
*A. salina* larvae after 48 h of exposure to F-UBE-HPC stained with acridine orange 100× (**d**), 200× (**b**,**c**,**e**,**f**), and 400× (**a**). (**a**–**c**)—blank; (**d**–**f**)—samples. The red fluorescence shows intracellular lysosomes activated in cell death processes (**e**,**f**); F-UBE-HPC—bioadhesive oral film loaded with *U. barbata* dry ethanol extract.

**Figure 9 pharmaceutics-14-01808-f009:**
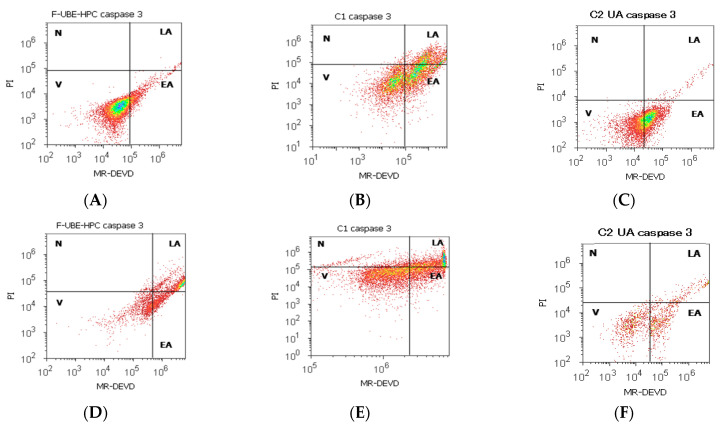
The activity of 3/7 effector caspases in normal blood cells (**A**–**C**) and CLS-354 tumor cells (**D**–**F**) after 24 h treatment with F-UBE-HPC; MR-DEVD patterns of F-UBE-HPC (**A**,**D**), 1% DMSO negative control (**B**,**E**); 125 µg/mL UA positive control (**C**,**F**). Statistical analysis of 3/7 caspases activity (**G**,**H**) in normal blood cell cultures (**G**) and CLS-354 tumor cell line (**H**), * *p* < 0.05 and ** *p* < 0.01 represent significant statistical differences between controls and sample made by paired samples *t*-test; V—viability; EA—early apoptosis; C1—negative control with 1% dimethyl sulfoxide (DMSO); C2UA—positive control with 125 µg/mL usnic acid (UA); F-UBE-HPC—bioadhesive oral film loaded with *U. barbata* dry ethanol extract.

**Figure 10 pharmaceutics-14-01808-f010:**
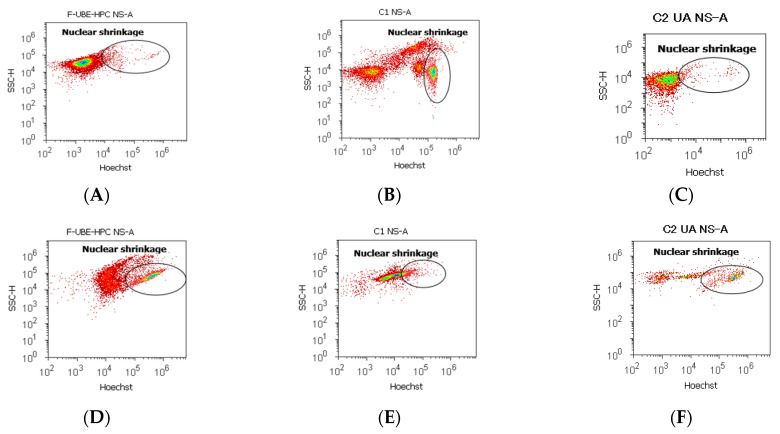
Nuclear shrinkage (**A**–**F**) and Lysosomal activity (**G**–**L**) after 24 h treatment with F-UBE-HPC in normal blood cells (**A**–**C**,**G**–**I**) and CLS-354 tumor cells (**D**–**F**,**J**–**L**). Hoechst patterns of F-UBE-HPC (**A**,**D**); 1% DMSO negative control (B, E); 125 µg/mL UA positive control (**C**,**F**); acridine orange patterns of F-UBE-HPC (**G**,**J**); 1% DMSO negative control (**H**,**K**); 125 µg/mL UA positive control (**I**,**L**). Statistical analysis of nuclear shrinkage and autophagy (**M**,**N**) in normal blood cell cultures (**M**) and CLS-354 tumor cell line (**N**). * *p* < 0.05 and ** *p* < 0.01 represent significant statistical differences between controls and sample made by paired samples *t*-test; NS—nuclear shrinkage; A—autophagy; C1—negative control with 1% dimethyl sulfoxide (DMSO); C2UA—positive control with 125 µg/mL usnic acid (UA); F-UBE-HPC—bioadhesive oral film loaded with *U. barbata* dry ethanol extract.

**Figure 11 pharmaceutics-14-01808-f011:**
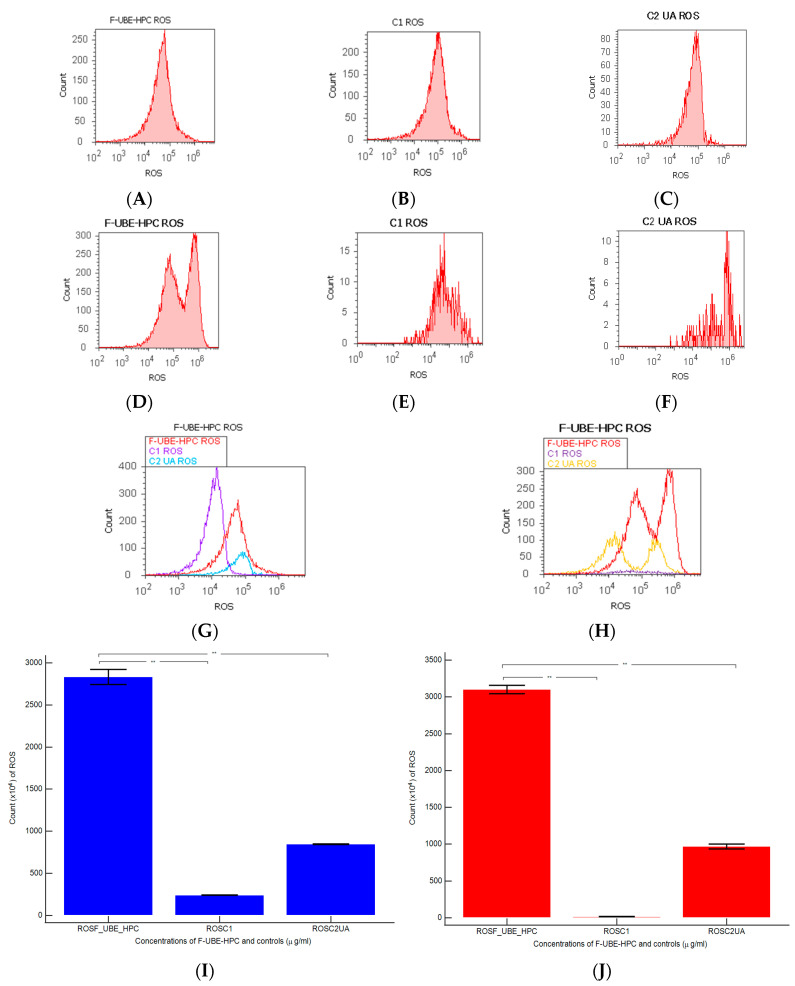
Reactive oxygen species (ROS) after 24 h treatment with F-UBE-HPC applied to normal blood cells (**A**–**C**) and CLS-354 tumor cells (**D**–**F**); 1% DMSO negative control (**B**,**E**); 125 µg/mL UA positive control (**C**,**F**); F-UBE-HPC and controls extrapolated on ROS axis (**G**,**H**); statistical analysis of ROS (**I**,**J**) in normal blood cell cultures (**I**); CLS-354 tumor cell line (**J**). ** *p* < 0.01 represents significant statistical differences between controls and samples made by paired samples *t*-test; C1—negative control with 1% dimethyl sulfoxide (DMSO); C2UA—positive control with 125 µg/mL usnic acid (UA); F-UBE-HPC—bioadhesive oral film loaded with *U. barbata* dry ethanol extract.

**Figure 12 pharmaceutics-14-01808-f012:**
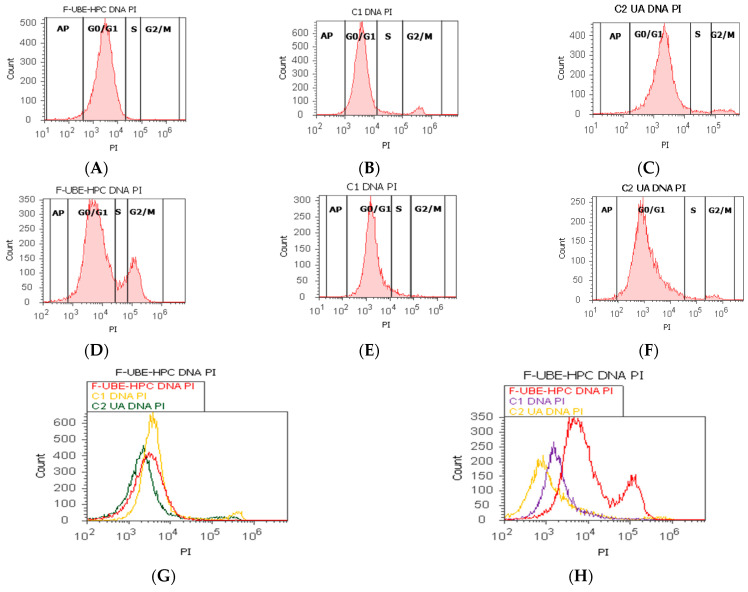
Cell cycle after 24 h treatment with F-UBE-HPC in normal blood cells (**A**–**C**) and CLS-354 tumor cells (**D**–**F**); PI/RNase patterns of F-UBE-HPC (**A**,**D**); 1%DMSO negative control (**B**,**E**); 125 µg/mL UA positive control (**C**,**F**); F-UBE-HPC and controls extrapolated on PI axis (**G**,**H**); statistical analysis of G0/G1, synthesis (S), and G2/M phases of the cell cycle (**I**,**J**) in normal blood cell cultures (**I**) and CLS-354 tumor cell line (**J**).* *p* < 0.05 and ** *p* < 0.01 represent significant statistical differences between controls and sample made by paired samples t-test; PI—propidium iodide; C1—negative control with 1% dimethyl sulfoxide (DMSO); C2UA—positive control with 125 µg/mL usnic acid (UA); F-UBE-HPC—bioadhesive oral film loaded with *U. barbata* dry ethanol extract; AP—apoptotic cell fraction (subG0/G1); S—synthesis of cell cycle phases.

**Figure 13 pharmaceutics-14-01808-f013:**
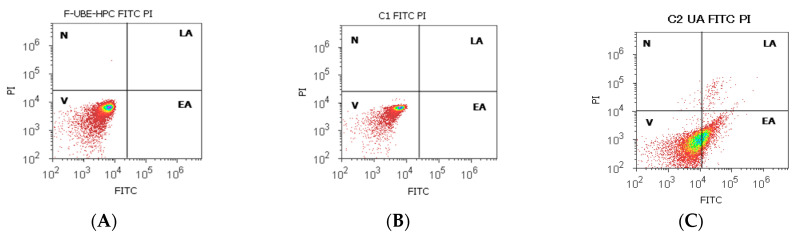
Cell apoptosis after 24 h treatment with F-UBE-HPC in normal blood cells (**A**–**C**) and CLS-354 tumor cells (**D**–**F**); annexin V-FITC/PI patterns of F-UBE-HPC (**A**,**D**); 1%DMSO negative control (**B**,**E**); 125 µg/mL UA positive control (**C**,**F**); statistical analysis of cell apoptosis (**G**,**H**) in normal blood cell cultures and (**B**) CLS-354 tumor cell line (**H**), *** p* < 0.01 represents significant statistical differences between controls and sample made by paired samples *t*-test; V—viability; EA—early apoptosis; C1—negative control with 1% dimethyl sulfoxide (DMSO); C2UA—positive control with 125 µg/mL usnic acid (UA); F-UBE-HPC—bioadhesive oral film loaded with *U. barbata* dry ethanol extract.

**Figure 14 pharmaceutics-14-01808-f014:**
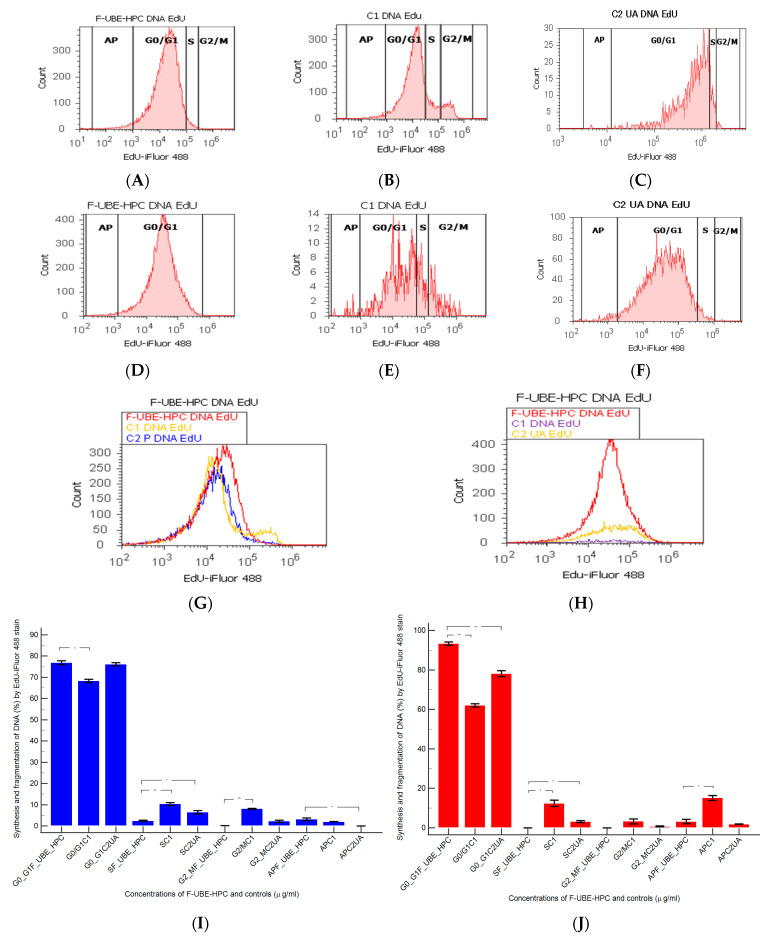
DNA synthesis (S) and DNA fragmentation (subG0/G1) after 24 h treatment with F-UBE-HPC in normal blood cell cultures (**A**–**C**) and CLS-354 tumor cells (**D**–**F**); EdU-iFluor 488 patterns of F-UBE-HPC (**A**,**D**); 1%DMSO negative control (**B**,**E**); 125 µg/mL UA positive control (**C**,**F**). F-UBE-HPC and controls extrapolated on EdU-iFluor 488 axis (**G**,**H**); statistical analysis of cell proliferation (**I**,**J**) in normal blood cell cultures and (**I**) CLS-354 tumor cell line (**J**). ** p* < 0.05 and ** *p* < 0.01 represent significant statistical differences between controls and sample made by paired samples *t*-test; C1—negative control with 1% dimethyl sulfoxide (DMSO); C2UA—positive control with 125 µg/mL usnic acid (UA). AP—apoptosis (subG0/G1); F-UBE-HPC—bioadhesive oral film loaded with *U. barbata* dry ethanol extract.

**Figure 15 pharmaceutics-14-01808-f015:**
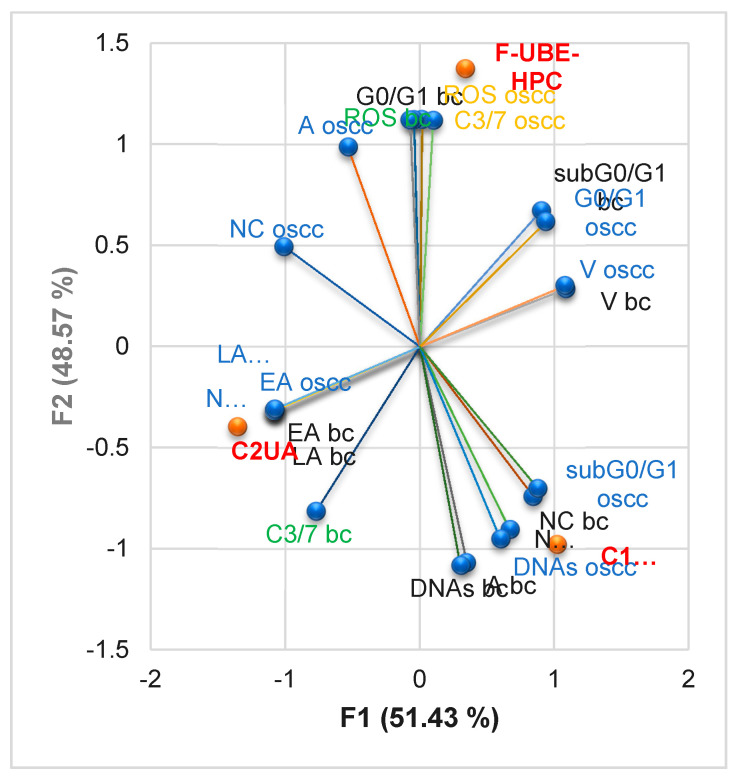
PCA-Correlation biplot between mechanisms (caspase 3/7 activity and cellular oxidative stress) and processes induced by F-UBE-HPC and both controls (C1-DMSO and C2UA) in normal blood cells (bc) and CLS-354 tumor cells (oscc). V—viability; EA—early apoptosis; LA—late apoptosis; N—necrosis; NC—nuclear condensation; A—autophagy; DNAs—DNA synthesis; subG0/G1—apoptotic cell fraction http://www.icms.qmul.ac.uk/flowcytometry/uses/apoptosis/dnafragmentation/ (accessed on 11 July 2022); G0/G1—cell cycle arrest in G0/G1; ROS—oxidative stress; C3/7—caspase 3/7 activity; F-UBE-HPC—bioadhesive oral film loaded with *U. barbata* dry ethanol extract.

**Table 1 pharmaceutics-14-01808-t001:** Bioadhesive oral film formulations.

Ingredients	Quantity (g)	
	F-UBE-HPC	R
UBE	0.30	-
Ethyl alcohol 96% (*v*/*v*)	10.00	10.00
PEG 400	5.00	5.00
HPC 20% water dispersion (*w*/*w*)	84.70	85.00

F-UBE-HPC—bioadhesive oral film loaded with *U. barbata* dry ethanol extract; R—bioadhesive oral film without active lichen extract.

**Table 2 pharmaceutics-14-01808-t002:** Thermal data of the studied samples.

Film	1st Step (%)	2nd Step (%)	3rd Step (%)
	TG (%)	TG(%)/T_max_ (°C)	TG(%)/T_max_ (°C)
R	2.4%	88.4%/357.6 °C	9.2%/468.3 °C
F-UBE-HPC	2.3%	85.6%/352.3 °C	12.1%/471.8 °C

R—bioadhesive oral film without active lichen extract; F-UBE-HPC—bioadhesive oral film loaded with *U. barbata* dry ethanol extract.

**Table 3 pharmaceutics-14-01808-t003:** Bioadhesive oral films’ pharmacotechnical characteristics.

Parameter *	Formulation Code	
	F-UBE-HPC	R
Weight uniformity (mg)	110 ± 4.77	107 ± 5.25
Thickness (mm)	0.093 ± 0.002	0.093 ± 0.003
Folding endurance value	>300	>300
Tensile strength (kg/mm^2^)	2.48 ± 1.58	2.57 ± 1.71
Elongation %	63.14 ± 1.94	62.75 ± 1.52
Moisture content % (*w*/*w*)	6.58 ± 0.44	6.23 ± 0.56
pH	7.10 ± 0.02	7.07 ± 0.01
Disintegration time (seconds)	118 ± 3.16	115 ± 4.19
Swelling ratio (% after 6 h)	289 ± 5.82	288 ± 6.13
Ex vivo biooadhesion time (minutes)	98 ± 3.58	98 ± 4.17

* Expressed as mean value ± SD; F-UBE-HPC—bioadhesive oral film loaded with *U. barbata* dry ethanol extract; R—bioadhesive oral film without active lichen extract.

**Table 4 pharmaceutics-14-01808-t004:** Microdilutions of Ceftriaxone, Terbinafine, and F-UBE-HPC films.

Microdilution	CTR		TRF 10.1 mg/mL	F-UBE-HPC 110 mg/mL
	30 mg/mL	122 mg/mL		
1	1.5	6.10	0.5	5.5
2	0.75	4.88	0.25	2.75
3	0.375	3.904	0.125	1.375
4	0.187	3.123	0.062	0.685
5	0.093	2.498	0.031	0.342
6	0.046	1.998	0.015	0.171
7	0.024	1599	0.007	0.085

CTR—Ceftriaxone, TRF—Terbinafine http://allie.dbcls.jp/pair/TRF;terbinafine.html (accessed on 5 July 2022), and F-UBE-HPC—bioadhesive oral film loaded with *U. barbata* dry ethanol extract.

**Table 5 pharmaceutics-14-01808-t005:** Inhibitory activity of F-UBE-HPC on *S. aureus* and *P. aeruginosa*.

Dil.	*S. aureus*	*P. aeruginosa*
CTR F-UBE	F-UBE-HPC	CTR	F-UBE-HPC
A	B	C	D	E	F	G	H
1		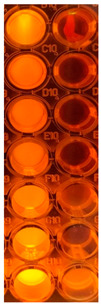		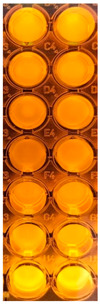		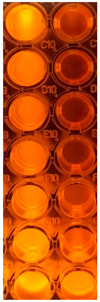	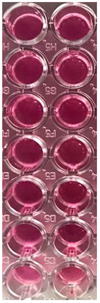	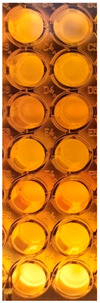
2
3
4
5
6
7
	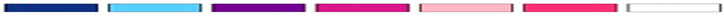 *

* Interpreting obtained results adapted from Madushan et al. [25] as blue—excellent; light blue—very good; violet—good; purple pink—moderate; light pink—low; pink—very low; white—no effect. CTR—Ceftriaxone and F-UBE-HPC—Bioadhesive film loaded with *U. barbata* dry ethanol extract; a,c,e,g well plates examined with a free eye; b,d,f,h well plates analyzed at 470 nm.

## Data Availability

All data are available in the manuscript.

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
