# Peer review of "Formulation and Development of Bioadhesive Oral Films Containing Usnea barbata (L.) F.H.Wigg Dry Ethanol Extract (F-UBE-HPC) with Antimicrobial and Anticancer Properties for Potential Use in Oral Cancer Complementary Therapy"

_pharmaceutics, 2022, doi:10.3390/pharmaceutics14091808_

Round 1

Reviewer 1 Report

This research described an oral film contained Usnea barbata (L.) dry ethanol extract for oral cancer therapy. This manuscript had detailed experiment methods and results to support conclusions. However, some result parts need more explanation. I suggest a revision before acceptance. Please refer to the following comments.

Major comments:

1.     Lack of drug release study. For example: using Frans diffusion cell for in vitro permeation test using animal tissue.

2.     In line 951-952, the author mentioned the F-UBE-HPC formulation showed fast release property. Please specify which study indicates the fast-release property. The film mucoadhesion study indicated the adhesive duration for the films was 98min. Please specify whether the loaded drug can be completely released out in this duration.  

3.     Why chose A. salina Larvae model for film cytotoxicity study?

4.     In in vitro study, please justify why selected Human blood cell as control?

5.     What the conclusion of the in vitro nucelar shrinkage study? Does the result indicate the developed formulation can cause negative impaction to normal blood cell?

6.     Please explain what the meaning of ROS level study (section 3.6.3).

7.     In 3.6.5 section, the apoptosis study, similar results were obtained in normal blood cell line and CLS-254 tumor cell line. Does it mean the developed formulation has no targeting (or specifying) property to cancer cell line?

Minor comments:

1.     I suggest simplifying abstract’s content.

2.     In Figure 3. Please indicate the difference between right and left column images. Please check the language of this figure’s caption.

3.     I suggest the authors to simplify it language in introduction and discussion parts and emphasize the significant of this work.

Reviewer 2 Report

The authors present the formulation, development and characterization of an oral bioadhesive film of HPC and PEG 400, loaded with ethanol extract of Usnea barbata. The researchers perform a physicochemical and pharmacotechnical characterization of the polymeric films. 

In addition, studies of antimicrobial activity and evaluation of anticancer activity were carried out.  

The manuscript presents a well-structured research, fulfilling the stated objectives. The methodology is well described, which would allow in an eventual case to reproduce the experimental trials. 

The results presented are excellent; the experimental quality of the data would guarantee a good discussion. 

It is advised to accept the manuscript for publication after some major changes.  

1. Although the research work shows a magnificent interdisciplinarity, reflected in the amount of results presented, the discussion of these results is limited, the authors dedicate more than half of the section to discuss the properties of the Usnea barbata extract, which is not the objective of the research. It would be expected to present a detailed discussion of the physicochemical and pharamcotechnical properties of the polymeric films, and to analyze in more detail the results of the biological tests, contrasting them with already published tests.   

2. In lines 951-952 it is mentioned that the release of the extract is rapid, however, no corroborating evidence is presented.  A release study would complete the research presented. 

Round 2

Reviewer 1 Report

The authors carefully answered previous comments and questions. The updated manuscript has reasonable revision. I recommend acceptance.

Reviewer 2 Report

The authors made the suggested corrections, I recommend publication of the manuscript.